# A Novel Graphene Oxide-Based Aptasensor for Amplified Fluorescent Detection of Aflatoxin M_1_ in Milk Powder

**DOI:** 10.3390/s19183840

**Published:** 2019-09-05

**Authors:** Xiaodong Guo, Fang Wen, Qinqin Qiao, Nan Zheng, Matthew Saive, Marie-Laure Fauconnier, Jiaqi Wang

**Affiliations:** 1Key Laboratory of Quality & Safety Control for Milk and Dairy Products of Ministry of Agriculture and Rural Affairs, Institute of Animal Sciences, Chinese Academy of Agricultural Sciences, 100193 Beijing, China (X.G.) (F.W.) (Q.Q.) (N.Z.); 2Chimie générale et organique, Gembloux Agro-Bio Tech, Université de Liège, Passage des Déportés 2, 5030 Gembloux, Belgium (M.S.) (M.-L.F.); 3Laboratory of Quality and Safety Risk Assessment for Dairy Products of Ministry of Agriculture and Rural Affairs, Institute of Animal Sciences, Chinese Academy of Agricultural Sciences, 100193 Beijing, China; 4State Key Laboratory of Animal Nutrition, Institute of Animal Science, Chinese Academy of Agricultural Sciences, 100193 Beijing, China

**Keywords:** aflatoxin M_1_, aptamer, graphene oxide, DNase I, food safety

## Abstract

In this paper, a rapid and sensitive fluorescent aptasensor for the detection of aflatoxin M_1_ (AFM_1_) in milk powder was developed. Graphene oxide (GO) was employed to quench the fluorescence of a carboxyfluorescein-labelled aptamer and protect the aptamer from nuclease cleavage. Upon the addition of AFM_1_, the formation of an AFM_1_/aptamer complex resulted in the aptamer detaching from the surface of GO, followed by the aptamer cleavage by DNase I and the release of the target AFM_1_ for a new cycle, which led to great signal amplification and high sensitivity. Under optimized conditions, the GO-based detection of the aptasensor exhibited a linear response to AFM_1_ levels in a dynamic range from 0.2 to 10 μg/kg, with a limit of detection (LOD) of 0.05 μg/kg. Moreover, the developed aptasensor showed a high specificity towards AFM_1_ without interference from other mycotoxins. In addition, the technique was successfully applied for the detection of AFM_1_ in infant milk powder samples. The aptasensor proposed here offers a promising technology for food safety monitoring and can be extended to various targets.

## 1. Introduction

Aflatoxin M_1_ (AFM_1_), one of the most toxic mycotoxins, was moved from group 2B to group 1 carcinogens by the International Agency for Research on Cancer (IARC) of the World Health Organization (WHO) [1,2]. AFM_1_ can be encountered in dairy products as a hydroxylate metabolite derived from feeding dairy cows aflatoxin B_1_-contaminated feeds [3,4,5]. Since dairy products are an important nutrient for humans, especially for infants, the presence of AFM_1_ in dairy products is one of the most serious hazards for food safety [6]. To protect humans from this health threat, many regulatory agencies have defined maximum residue levels (MRLs) for AFM_1_ in dairy products [7,8]. In Brazil, China, and USA, the maximum level of AFM_1_ in milk has been fixed to 0.5 μg/kg [9,10]. The European Commission Regulation has set much more restrictive limits, i.e., 0.05 μg/kg in milk products for adults, and this level is lowered to 0.025 μg/kg for baby and infant products [11]. Considering the severe toxicity and low permitted level of AFM_1_, a simple, rapid and inexpensive diagnostics with high sensitivity and specificity is vitally required for food safety.

Analytical methods such as high-performance liquid chromatography (HPLC) with fluorescent detectors [12,13,14] and high-performance liquid chromatography coupled with mass spectrometric detectors [15,16] have been developed for quantitative detection of AFM_1_. All these procedures rely on expensive instruments, qualified staff and complicated time-consuming pretreatments. Meanwhile, enzyme-linked immune sorbent assays (ELISA) [17,18,19] have gained great attention for AFM_1_ analysis owing to the advantages of rapid, low-cost and high-throughput application. However, expensive, time-consuming and laborious antibody production and antibody instability during storage limit the practical applications of these assays. Therefore, the design of a simple, cheap and sensitive method for rapid detection of AFM_1_ has become a research hotspot.

Aptamers, single-stranded DNA or RNA oligonucleotides, have been widely applied for recognizing targets such as proteins, nucleic, cells, tissues and small molecules with strong affinity and high specificity similar or even superior to those of antibodies [20,21]. Up to now, a series of aptamer-based biosensors for the detection of several mycotoxins including ochratoxin A (OTA), aflatoxin B_1_ (AFB_1)_ and AFM_1_ have been developed [22,23,24,25,26,27]. In these homogeneous methods, the recognition reaction between the aptamer and the target is based on single-site binding, which might limit the method sensitivity. Thus, the development of aptasensors coupled with signal amplification strategies for AFM_1_ detection is an on-going challenge. In our previous study [27], a sensitive aptasensor based on real-time quantitative polymerase chain reaction (RT-qPCR) for AFM_1_ was developed. However, the preparation of the qPCR-based aptasensor requires a tedious and time-consuming process with long incubation periods in rigorous conditions. Therefore, there is a demand for aptasensors that could be applied for rapid and real-time analysis of AFM_1_.

Graphene oxide (GO), a two-dimensional nanomaterial, is a very promising tool for the construction of biosensors because of its extraordinary electrical, thermal and mechanical properties [28,29,30,31]. The predominantly distance-dependent fluorescence quenching ability makes GO a highly efficient fluorescence quencher [32,33,34]. In addition, previous research demonstrated that single-stranded DNA can significantly interact with GO through π stacking between DNA bases and hexagonal cells of GO [35]. Importantly, GO could protect DNA aptamers from nuclease cleavage as a result of the hydrophobic stacking interactions between nucleobases and GO [36,37,38]. To the best of our knowledge, graphene oxide-based nuclease signal amplification aptasensors for AFM_1_ determination have not been reported.

In this study, a new graphene oxide-based aptasensor for the specific detection of AFM_1_ was developed, which combined the ability of GO to protect the aptamer from nuclease cleavage with that of DNase I to cleave the aptamer for a target cycling signal amplification. The presence of AFM_1_ induced the release of the aptamer from the surface of GO because of the formation of an AFM_1_/aptamer complex, which resulted in the cleavage of the aptamer by DNase I and the release of AFM_1_ for a new cycle. Therefore, a cycling signal amplification was achieved to improve detection sensitivity. A good linear relationship was measured between the change of the fluorescence intensity signal and AFM_1_ levels. 

## 2. Experimental

### 2.1. Materials and Reagents

AFM_1_ was purchased from Sigma-Aldrich (USA). AFB_1_ was obtained from the National Standard Reference Center (Beijing, China). OTA, zearalenone (ZEA) and α-zearalenin (α-ZOL) were purchased from Pribolab Co. Ltd (Singapore). Graphene oxide was purchased from Sangon Biotechnology Co. Ltd. (Shanghai, China). DNase I (RNase-free) was obtained from Takara Bio Co. Ltd. (Dalian, China). Other chemicals such as anhydrous calcium chloride (CaCl_2_), sodium chloride (NaCl), potassium chloride (KCl) and 2-Amino-2-(hydroxymethyl)-1,3-propanediol (Tris) were purchased from Shanghai Chemical Reagent Company (Shanghai, China). All other chemicals were analytical-grade and were used as received without further purification. Water was purified with a Milli-Q purification system. DNA oligonucleotides were chemically synthesized by Sangon Biotechnology Co. Ltd. (Shanghai, China) and purified by HPLC. The sequence of the AFM_1_ aptamer was optimized according to our previous study [27] and was modified by FAM (carboxyfluorescein). The sequence of the FAM-labelled AFM_1_ aptamer was as follows:

5′5′-FAM–ATCCGTCACACCTGCTCTGACGCTGGGGTCGACCCG-3′

### 2.2. Fluorescent Response of the Amplified Aptasensor for AFM_1_

In this amplification strategy, the FAM-labelled AFM_1_ aptamer was diluted to 200 nM in Tris buffer (10 mM Tris, 120 mM NaCl, 5 mM KCl, 20 mM CaCl_2_, pH 7.0), and 20 μg mL^−1^ of GO was added to the working solution for 15 min at room temperature to form the aptamer/GO complex and quench the fluorescence. Subsequently, solutions at different concentrations of AFM_1_ and DNase I (200 U) were simultaneously added to the aptamer/GO solutions, and the mixtures were and incubated at room temperature for 1 h. Afterwards, the fluorescence intensities of the mixtures were recorded using an F-7000 fluorophotometer (Hitachi, Tokyo, Japan). The emission spectra were measured in the range of 510 to 630 nm with the excitation wavelength at 480 nm, and slit widths for both of the excitation and the emission were set at 10 nm.

### 2.3. Specificity Analysis 

To evaluate the specificity of this aptasensor for AFM_1_ over other mycotoxins, four different mycotoxins (AFB_1_, OTA, ZEA and α-ZOL) were measured at the same concentration of 4 ng mL^−1^. The other experimental procedures were the same as AFM_1_ determination, and the changes of fluorescence intensity for these mycotoxins were compared.

### 2.4. Method Validation

The feasibility and practicability of this sensing platform was verified by the quantitative detection of AFM_1_ in infant milk powder samples. The samples were spiked with AFM_1_ at 0, 1.5, 2.5 and 5 μg/kg (three replicates per treatment). Each sample was accurately weighed (0.5 g) into 10 mL centrifuge tubes. Then, 2.5 mL of extraction solution (70% methanol in water) was added to extract AFM_1_ from the samples. The entire mixture was vortexed using Vortex-Genie 2 (Scientific Industries, Bohemia, NY, USA) for 5 min and then centrifuged at 10,000 g for 10 min. The supernatant was obtained and concentrated to 0.5 mL under a nitrogen stream. Subsequently, each residue was re-dissolved in 2 mL of aqueous methanol solution (5% methanol in water). Finally, the extracts were measured by the fluorescence signal amplification experiment.

### 2.5. Statistical Analysis

Fluorescence-emission spectra curves for AFM_1_ were plotted using Origin 8.0 software (OriginLab Corporation, Northampton, MA, USA). Linear regression analysis of the fluorescence intensity as a function of the concentration of AFM_1_ was carried out with Microsoft Excel. Each analysis including AFM_1_ calibration curve standards and test samples was performed in triplicate. Standard deviations (SDs) and means of fluorescence intensity were determined from three replicates.

## 3. Results and Discussion 

### 3.1. Design Strategy for AFM_1_ Detection Based on a Graphene Oxide Sensing Platform

GO has many advantages due to its unique properties, including its great binding ability to single-stranded DNA (such as aptamers) through π stacking interactions between nucleobases and GO nanosheets and its high distance-dependent fluorescence quenching performance [35,39]. A GO-based aptasensor for the detection of AFM_1_ was developed taking advantage of the above properties. A schematic illustration of the sensing platform is presented in Figure 1. In this sensing method, when the FAM-modified aptamer was incubated with the GO solution, the fluorescence signal quenched dramatically, demonstrating a strong binding between the aptamer and GO, with a high quenching efficiency. Upon the addition of AFM_1_, an AFM_1_/aptamer complex formed. Such an interaction can lead to a conformational change in the aptamer, causing a separation of the conjugated aptamer from the surface of GO. Thus, fluorescence is recovered, since GO would not be able to quench the fluorescence efficiently owing to the long distance. In order to confirm that the presence of AFM_1_ can lead to the formation of an AFM_1_/aptamer complex and subsequently to fluorescence recovery, 10 ng mL^−1^ of AFM_1_ was added to a Tris buffer solution that contained 200 nM of AFM_1_ aptamer and 20 μg mL^−1^ of GO. As seen in Figure 2, a significant fluorescence enhancement was observed, demonstrating that the AFM_1_/aptamer complex was formed. More importantly, the covalently modified FAM had no impact on the recognition ability of the AFM_1_ aptamer.

DNase I was adopted as a signal amplification strategy to improve the sensitivity of the aptasensor. As shown in Figure 1, upon the addition of AFM_1_ and DNase I, the formation of the AFM_1_/aptamer complex caused the dissociation of the aptamer conjugate from GO, and subsequently the aptamer was digested by DNase I. Once AFM_1_ was released from the AFM_1_/aptamer complex, it was again available to bind to another aptamer, inducing a cyclic amplification of the fluorescence signal. As a consequence, a strong amplification of the fluorescence signal could be achieved for the quantification of AFM_1_.

### 3.2. Optimization of the Experimental Conditions

The concentration of GO would influence the fluorescence quenching efficiency. Therefore, to optimize the sensing platform, the effect of GO concentration on the change of the fluorescence signal was investigated. Various concentrations of GO were added to a solution containing 200 nM of AFM_1_ aptamer. As seen in Appendix A, the fluorescence intensity decreased with increasing amounts of GO and reached the lowest level at a concentration of GO of 20 μg mL^−1^. Thus, 20 μg mL^−1^ of GO solution was used for further sensing experiments.

To improve the signal amplification efficiency, the optimization of the concentration of DNase I was essential. In this experiment, we measured the fluorescence intensity of the complex with 10 ng mL^−1^ of AFM_1_. Various amounts of DNase I were added to the GO/aptamer solution containing 200 nM of AFM_1_ aptamer and 20 μg mL^−1^ of GO. As seen in Appendix A, the fluorescence intensity increased as the DNase I concentration increased from 0 to 200 U, and the highest level of fluorescence was observed at 200 U of DNase I. In this case, the optimal amount of DNase I was determined as 200 U.

### 3.3. Analytical Performance of the Aptasensor

Under optimal conditions, the signal responses of the GO-based aptasensor to different concentrations of AFM_1_ were measured using a DNase I-induced target recycling amplification platform. The fluorescence emission spectrum was measured for AFM_1_ determination with excitation and emission wavelengths of 480 nm and 520 nm, respectively. As seen in Figure 3, the fluorescence intensity increased as the concentration of AFM_1_ increased from 0.2 to 10 μg/kg. The calibration curve of fluorescence intensity versus AFM_1_ concentrations was linear, as *F* = 65.77 *C* + 46.334 (R^2^ = 0.9939), in which *F* is the fluorescence intensity, and *C* is AFM_1_ concentration. The limit of detection of the amplification aptasensor was determined to be 0.05 μg/kg, which was calculated at the signal-to-noise ratio of 3. As shown in Table 1, the aptasensor displayed a sensitivity for AFM_1_ comparable to those of other instrumental and rapid screening methods reported previously.

### 3.4. The Specificity of the Aptsensor

The specificity of the aptasensor was also investigated to assess the effect of other mycotoxins. The change of fluorescence intensity was measured under experiment conditions identical to those used for AFM_1_ detection in the presence of four other mycotoxins (AFB_1_, OTA, ZEA and α-ZOL) at a concentration of 4 ng mL^−1^. It can be seen that significantly higher fluorescence intensity was obtained in the case of AFM_1_ determination in comparison with other mycotoxins and the control (Figure 4), which indicated that the specificity of this amplifying sensing platform is high for AFM_1_ determination.

### 3.5. Method Validation

Ultimately, the applicability and reliability of the aptasensor platform were evaluated by detecting different concentrations of AFM_1_ in infant milk powder samples. As indicated in Table 2, the recovery of the spiked infant milk powder samples ranged from 92% to 126%, demonstrating that the amplification strategy developed in this work can be useful as a quantitative method for AFM_1_ analysis in real samples for food safety.

## 4. Conclusions 

A novel graphene oxide-based aptasensor was developed for the detection of AFM_1_ with high sensitivity and specificity. This technique uses the properties of GO as an aptamer protector against nuclease cleavage, thereby allowing DNase I to cleave the aptamer for a target cycling signal amplification. Under the optimal conditions, a good linear relationship was detected between fluorescence intensity and AFM_1_ levels in the range of 0.2 to 10 μg/kg, with a detection limit of 0.05 μg/kg. Satisfactory recoveries were measured in infant milk powder samples spiked with different concentrations of AFM_1_. Furthermore, the aptasensor proposed in this work is rapid, simple and low-cost in comparison with other methods reported previously. This study could thus provide a very promising platform for the analysis of AFM_1_ in dairy products. More importantly, the aptasensor could be improved by replacing aptamer sequences for the detection of other food safety targets.

## Figures and Tables

**Figure 1 sensors-19-03840-f001:**
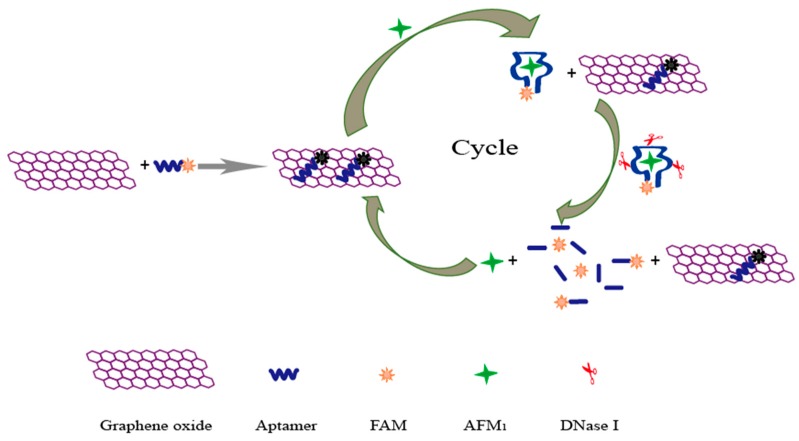
Illustration of the aptasensor for the detection of aflatoxin M_1_ (AFM_1_). FAM: carboxyfluorescein.

**Figure 2 sensors-19-03840-f002:**
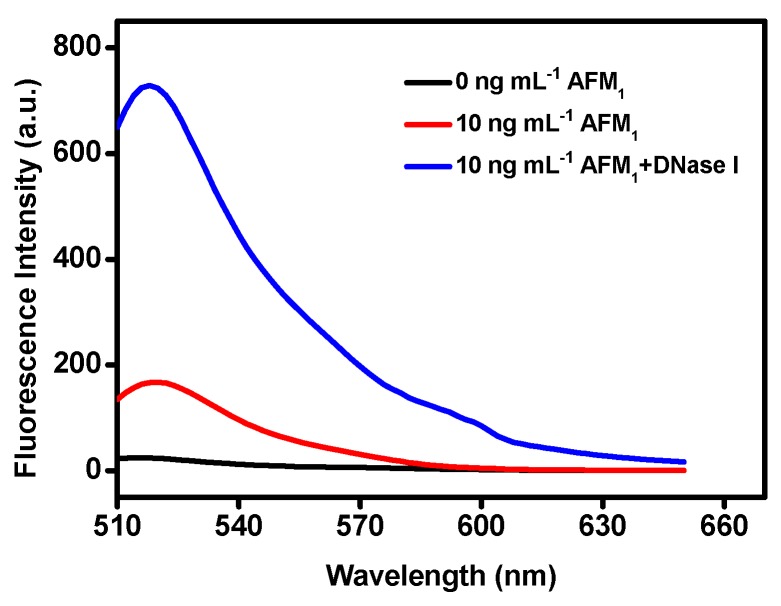
Fluorescence emission spectra of the aptasensor in the absence (0) of AFM_1_ and in the presence of 10 ng mL^−1^ AFM_1_ or 10 ng mL^−1^ AFM_1_ and 200 U DNase I. The excitation wavelength (λ_ex_) was set at 480 nm. Conditions: 200 nM AFM_1_ aptamer, 20 μg mL^−1^ graphene oxide (GO) in Tris buffer (10 mM Tris, 120 mM NaCl, 5 mM KCl, 20 mM CaCl_2_, pH 7.0).

**Figure 3 sensors-19-03840-f003:**
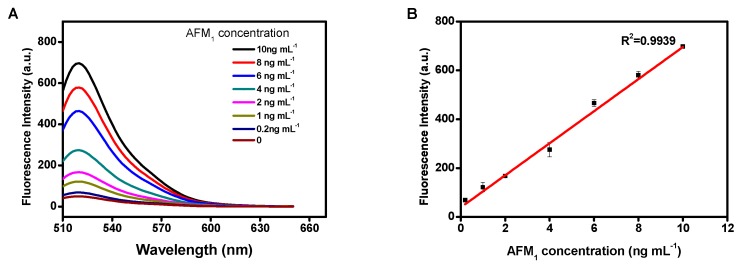
(**A**) Fluorescence emission spectra of the aptasensor with the addition of AFM_1_ at various concentrations. (**B**) Linear relationship between fluorescence intensity and AFM_1_ concentrations in the range of 0.2 to 10 ng mL^−1^.

**Figure 4 sensors-19-03840-f004:**
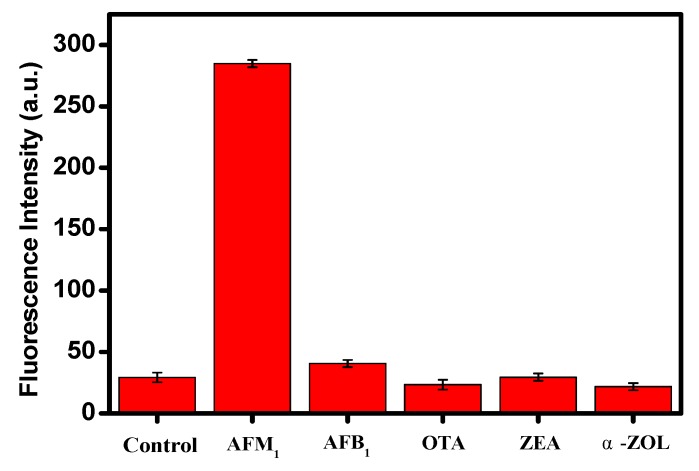
Fluorescence intensity in the absence (control) and presence of 4 ng mL^−1^ of the mycotoxins AFM_1_, AFB_1_, ochratoxin A (OTA), zearalenone (ZEA) and α-zearalenin (α-ZOL). The experiment conditions were as follows: Excitation wavelength (λ_ex_) at 480 nm, 200 nM of AFM_1_ aptamer, 20 μg mL^−1^ of GO, 200 U of DNase I. Every data point is the mean of three replicates.

**Table 1 sensors-19-03840-t001:** Comparison of the sensitivity of currently available methods for the detection of AFM_1_.

No.	Method	LOD	Reference
1	Fluorometric Sensor	0.05 µg L^−1^	[40]
2	Electrochemical Immunosensors	0.001 µg L^−1^	[41]
3	Indirect Competitive ELISA	0.04 µg L^−1^	[3]
4	Impedimetric Biosensor	1 µg L^−1^	[42]
5	HPLC	0.026 µg kg^−1^	[12]
6	Cellular Biosensor	0.005 µg L^−1^	[43]
7	Direct Chemiluminescent ELISA	1 ng L^−1^	[44]
8	DART-MS	0.1 µg kg^−1^	[9]
9	SPE–UPLC–MS/MS	1.5 ng kg^−1^	[16]
10	Impedimetric Aptasensor	1.15 ng L^−1^	[10]
11	Graphene Oxide-based Aptasensor	0.05 µg kg^−1^	This work

**Table 2 sensors-19-03840-t002:** Determination of AFM_1_ spiked into infant milk powder samples.

Sample	Spiked Concentration (μg/kg)	Detected Concentrations Mean^a^ ± SD ^b^ (μg/kg)	Recovery (%)
Infant Milk Powder	0	ND ^c^	-
	1.5	1.48 ± 0.06	98
	2.5	2.3 ± 0.42	92
	5.0	6.3 ± 0.06	126

^a^ The mean of three replicates; ^b^ SD = standard deviation; ^c^ ND = not detected.

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
