# Peer review of "A Novel Graphene Oxide-Based Aptasensor for Amplified Fluorescent Detection of Aflatoxin M1 in Milk Powder"

_sensors, 2019, doi:10.3390/s19183840_

Round 1

Reviewer 1 Report

Reviewer’s Comments:

This manuscript showed the method for signal amplification of graphene-oxide based aptasensor. However, the reviewer suggests the following points can be revised for the improvement of the manuscript.

Decision: Minor revision

In Figure 1, the reviewer barely understood with the text. This figure is not intuitive. The reviewer thought the reason is AFM was not shown with the digested aptamer. It looks like more AFM was added. This article is very logical, and the data is very interesting. However, the reviewer became curious about the mention on page 2, in line 58-59. Some cases, not all, which reviewer was participated and not published yet, were shown that target didn’t bind to aptamer with 1:1 binding ratio. So, if the author can prove, the reviewer thinks it would be better to do it.

The reviewer suggests some revision of the above-mentioned points in the manuscript.

Author Response

Thanks for your attention and review.

Reviewer 2 Report

This manuscript reports the graphene oxide-based aptasensor for the detection of aflatoxin M1. Authors lowered the detection limit of aflatoxin to 0.05ug/kg and succeeded in the detection of toxin in milk powder. The experimental methods are well explained and the manuscript is also well written thoroughly. Therefore, I recommend that this manuscript is publishable in Sensors.

Author Response

Thanks for your kind review and attention.

Reviewer 3 Report

The article introduces a graphene oxide-based (GO) aptasensor for the detection of aflatoxin M1 in milk powder. The binding event is detected by a fluorescence signal. GO is able to quench the fluorescence signal of the aptamer as well as to protect it from DNase dependent degradation. After ligand binding, the aptamer separates from GO and is degraded via DNase, resulting in a clear fluorescence signal. In this way, AFM1 could be specifically detected in milk powder samples, since the aptamer can discriminate between chemically related ligands.

The topic is very interesting and the functionality of the sensor is clearly explained. The necessity of such sensors with regard to food control is made clear so that the experiments carried out appear comprehensible and meaningful. In addition, the article is written very clearly and well understandable. Nevertheless, there are some minor remarks: 

Line 23: You wrote after DNase dependet degradation of the aptamer AFM1 is set free and is available for the next cycle. Does this mean that the ligand is detected several times? Maybe this falsifies the later results, because in this way, a "higher" concentration is ultimately measured, since the ligand is detected several times.

Line 54: Beginning of sentence is not correct

Line 96: You wrote „.55'“ - is that right?

Line 101 and 110: I’m missing more precise information on concentrations. Large volume changes lead to changes in concentrations used (especially Tris; Aptamer; GO). Is the aptamer concentration final 200 nM or at the beginning?

Line 145: You wrote, that FAM has no influence on the binding affinity à no reference

Line 148: Figure 2
According to Reference 36, 37, 38 (line 71), GO protects against degradation by DNase1. A sample with 0 ng/ml AFM1 + DNase1 could confirm this.

Line 185: Figure 3
You measured from 510 nm - 650 nm. Why? The peaks are located at the outer left edge of the figure and are not completely visible.  Why? What is before the peak? Why do you show so much plateau phase at the end?

Line 189: Table 1
LOD with uniform units (like ppb or ng/ml) simplifies comparison

Line 198: Figure 4.
Comparison with Fig. 3: 4 ng/ml leads only to a small increase in fluorescence. All other measurements were performed with 10 ng/ml. Why is  this measurement with only 4 ng/ml? What is the control?

Line 210: Table 2
Recovery: 92 - 126%. Where do the increased values come from? Here I suspect a connection to line 23: Ligand is measured several times and higher values are achieved.

Line 222: Here dairy products are addressed. Only milk powder was tested.  What about fresh milk, yoghurt, cheese, etc.? Did you already check the functionality of the aptasensor in samples like these?

Line XX: The Supplement was not uploaded and cannot be reviewed.

Author Response

Thanks for your kind suggestions and attention.
